# Physical Sciences Teachers' Enactment of Simulations in 5E Inquiry-Based Science Teaching

Gloria Makamu and Umesh Ramnarain *

Department of Science and Technology Education, University of Johannesburg, Johannesburg 2006, South Africa
* Correspondence: uramnarain@uj.ac.za

**Abstract:** The aim of this study was to investigate the pedagogical actions of Physical Sciences teachers when enacting simulations in 5E inquiry-based science teaching for current electricity. Three grade 10 teachers from three high schools who were teaching at schools where ICT resources are available participated in this study. Data was collected by means of lesson observations and interviews. The lesson observation and interview transcripts were coded to generate themes. The results of this study showed that the simulations enable teachers to afford learners with opportunities to engage in hands-on inquiry based on the 5E model. The hands-on activities that students engage in on the simulation help them to explain phenomena from evidence and also allow them to acquire autonomy from the teacher. When students are hands-on, they get the chance to test their hypothesis and also to develop their understanding of the phenomenon that is investigated. Through the use of the simulated activity, teachers were able to support leaners to reflect on activities to reconcile their new knowledge with previous ideas. While it is acknowledged that PhET simulated activity is not a substitute for hands-on practical work in a laboratory, the findings show it can be a powerful tool for supporting inquiry learning.

**Keywords:** 5E inquiry; simulations; physics; constructivist approach

## 1. Introduction

In South Africa, the curriculum goal of inquiry-based learning is underlined in Specific Aim 2 of the Curriculum Assessment Policy Statement (CAPS), where it requires learners in their immediate environment to explore objects, situations and events, to collect data and record information, and to accurately draw conclusions [1]. From a teaching perspective, the complex scientific process of inquiry can be divided into smaller, logically connected units called inquiry phases, that support learners and draw attention to important features of scientific thinking [2]. One of these instructional models based on phases is the 5E inquiry model, which defines five phases. These phases are: Engagement, Exploration, Explanation, Elaboration and Evaluation [3]. Ruiz-Martin and Bybee [4] stipulate that utilising a 5E instructional model in the classroom helps to facilitate inquiry-based learning because it focuses on constructivist principles, and it emphasises the explanation and investigation of a phenomena. The model also emphasises the importance of evidence in supporting claims and experimental design [5].

Inquiry-based learning (IBL) is a constructivist approach where students have control over their learning process and they are provided with an opportunity to explore, discover, construct knowledge, understand concepts, think critically and reflect, instead of relying on teacher direction [6]. The value of inquiry-based learning is explained in terms of it being a constructivist, inductive way of active learning that emphasises questioning, data analysis and critical thinking to create meaning and knowledge, in a real learning environment [7]. However, a number of challenges have been identified in implementing an inquiry-based practice. Some of the challenges that are cited in South Africa include a lack of teaching time, the inadequacy of resources, large classes, and security issues [8]. Therefore, the use

of simulations could provide an alternative to traditional laboratory experimentation that has been constrained due to the aforementioned factors.

Within the South African education landscape, the use of ICT has been hailed as a panacea to poor achievement in science subjects [9] and is strongly encouraged by the education ministry [1]. In schools, simulations enable children to engage with powerful ideas and conduct explorations that are not usually possible in classrooms [10]. Podolefsky et al. [11] says that computer simulations provide instant feedback on the results of a virtual 'experiment' as well as opportunities for rich and dynamic educational experiences for students. Khan [12] defines interactive computer simulations as "a computer program that attempts to simulate a model of a particular system" (p. 216). An example of a simulation commonly used in this country is the "PhET simulation" which is an acronym for Physics Education Technology. These simulations are freely available on the Internet, and therefore readily accessible. They can be used by Physical Sciences and Mathematics teachers to encourage scientific inquiry. PhET simulations (sims) are 'animated, interactive, and game-like environments in which students learn through exploration' [13]. These simulations put emphasis on the connections between real-life phenomena and the fundamental science [14]. The visual and conceptual models of Physical Science specialists are made accessible to students through the PhET simulations [14].

A number of studies found that PhET simulations can replace real laboratory equipment in physics courses [15,16]. Interactive simulations help students to observe invisible phenomena beyond the range of the naked eye, such as atomic and molecular scale processes, as well as allowing visual representation of non-physical concepts such as magnetic field lines [17]. According to Perkins et al. [13], PhET simulations can be used for different purposes, such as doing research, lecturing, conducting inquiry group activities, doing homework, and lab activities. These sims can be used for introducing a new topic, construction of concepts and skills, reinforcement of ideas, and to help students reflect on what they have learnt [18].

Despite the widely reported benefits of simulations, few studies provide insights into the possible roles of the teacher pedagogical strategies employed in the use of simulations [12]. This study will investigate the pedagogy of science teachers in teaching topics with Physics Education Technology (PhET) computer simulations. The research was guided by the following question: What are the pedagogical actions of Physical Sciences teachers when enacting simulations in 5E inquiry-based science teaching for current electricity? Studies on learners' conceptions of electricity have shown that this is a content area that is strewn with misconceptions, due to the abstract nature of electric circuits [19–21]. Some of the common misconceptions that have been identified include current decreases as it travels around the circuit; the current is shared equally by all devices in a circuit; the further the light bulb is from the power source (battery), the dimmer the light bulb; and the more resistors that are added in parallel, the greater the total resistance.

## 2. The 5E Instructional Model of Inquiry

This research was framed by the 5E instructional model of inquiry. The 5E instructional model of inquiry is a technique that science teachers use to produce scientifically literate students [22]. This model of instruction was developed under the Biological Science Curriculum Study (BSCS) project in the United States. According to Chitman-Booker and Kopp [22], the 5E instructional model provides a framework for teachers to develop students' understanding of scientific ideas and concepts. It also 'engages students' thinking, and then allows for explorative discovery and factual learning to deepen student understanding of content matter' [22].

The 5E instructional model can be used to design a science lesson, and it is based upon cognitive psychology, constructivist learning theory, and best practices in science teaching [23]. This research adopted a 5E instructional model because using a learning cycle approach in the classroom supports the facilitation of inquiry practices that concentrates on constructivist principles and emphasizes the explanation and investigation of

phenomena, the use of evidence to support conclusions, and experimental design [23]. The 5E instructional model is also important for this study because Bybee [24] states that 'using this approach, students redefine, reorganize, elaborate, and change their initial concepts through self-reflection and interaction with their peers and their environment. Learners interpret objects and phenomena and internalize those interpretations in terms of their current conceptual understanding' (p. 176). Some of these initial conceptions are misconceptions that act as a barrier to learning science. Research that has been conducted on the use of the 5E model in the teaching of electric circuits has revealed improvement in learners' achievement. For example, a study by Guzel [25] showed that the 5E model provided a better understanding by students, increased the motivation related to the lesson, and created a positive effect on understanding abstract concepts in electricity.

To remediate the misconceptions of students, Sari et al. [26] used the 5E model with simulations. Their findings showed that in students with alternate conceptions, the incorporation of simulations into the 5E learning model would produce considerable conceptual change. The 5E learning cycle is designed to guide teachers to select, plan, and sequence teaching to promote learning outcomes [27]. This model requires the instruction of five discrete elements that are reflected in Figure 1 below.

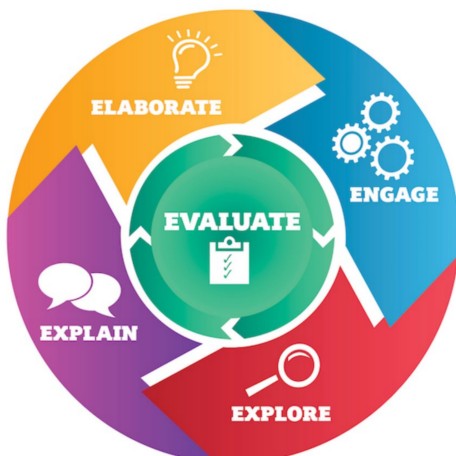

**Figure 1.** The 5E model of inquiry-based teaching and learning [28].

In the engagement, the teacher stimulates the students' interest and curiosity [29], and draws their attention through demonstration or asking learners a question to check the learners' prior knowledge, or any misconceptions that the learners might hold [30]. The engaged phase is centerd on identifying the prior ideas that are then investigated in the subsequent phases of the 5E. In this phase, students are given activities which help them to focus on the learning task and it also introduces them to a new problem they have to solve [3].

During the explore phase, students establish relationships, develop patterns, and question the process through the exploration of objects, events, or scientific phenomena [3]. Exploration experiences provide a common basis of activities for students to recognize existing ideas (or misconceptions), processes and skills, and to promote conceptual change.

The explain phase focuses the attention of students on a specific aspect of their experiences of engagement and discovery and offers opportunities to show their conceptual understanding, process skills, or behaviors process [3]. This phase often gives teachers opportunities to introduce a concept, method, or skill directly. Learners clarify their comprehension of the concept. The teacher supports students in acquiring deeper understanding, which is a vital component of this process [3].

In the elaboration phase teachers challenge the conceptual understanding and skills of students and extend them [3]. The students gain deeper and broader comprehension, more

information, and appropriate skills through new experiences. By performing additional exercises, students apply their understanding of the concept [3].

The evaluation phase evaluates students' understanding and abilities and provides teachers with opportunities to measure student progress towards the achievement of educational goals [3].

While the 5E model has been expanded to the 7E model (Engage/Elicit, Explore, Explain, Elaborate/Extent, Evaluate) by Eisenkraft [31], within the South African context teachers are most familiar with the 5E model, with research also being concentrated on the use of the 5E model.

## 3. Method

This study employed a qualitative case study design that enabled the researchers to generate an in-depth, multi-faceted understanding of a complex issue in its real-life context [32]. The focus of this inquiry was to describe and explain teacher use of simulations in 5E inquiry-based science teaching for current electricity.

### 3.1. Sampling

Purposeful sampling was employed in carrying out this study. Purposeful sampling is normally used in a qualitative study for the identification of information-rich cases. In this study, the main criterion in the selection of the sample were that the teachers were supportive of an inquiry-based pedagogy and recognized the use of ICT in science teaching. A further criterion was that the teachers were teaching at schools where ICT resources, such as tablets, were available. The sample comprised three Physical Sciences teachers from three high schools in Gauteng province, South Africa. These schools were also readily accessible to the research in terms of proximity to where the researcher was located. All South African public schools are categorized into five groups, called quintiles, largely for purposes of the allocation of financial resources. Quintile one is the 'poorest' quintile, while quintile five is the 'least poor' and is well resourced [33].

### 3.2. Profile of Teachers and Learners

Three teachers (Peter, Susan, and Thabo) participated in this study.

Peter is teaching in a school located in the Gauteng province and it is classified under quintile five. The school has a population of 1151 learners. He is 34 years old. It is a well-resourced school, and the learners use tablets and computers. The learners are using the miEbooks app for accessing their books. miEbooks is an app where the learners' textbooks, videos and slides are uploaded. All the classrooms have whiteboards and data projectors. Teachers use tablets or laptops to present their lessons. This school was founded in 1917. Peter is currently doing a Masters in Chemical Engineering and he holds an Honours degree in chemical engineering. He is a Physical Sciences, Natural Sciences, and Mathematics teacher. He has seven years' teaching experience and is originally from Zimbabwe. Peter's Grade 10 class had 35 learners, with 20 girls and 15 boys.

Susan is also teaching in Gauteng province. She is 42 years old. The school is a quintile five school with a population of 1012 learners. The learners have tablets and smart phones. The teacher uses a smartboard, laptop, and a computer to access computer simulations. This is a well-resourced school. Susan is currently doing her PhD in Science Education at Wits University. Her highest qualification is Masters in Science Education, which she obtained in 2017. She has been teaching for 20 years and the subjects she taught in these years are Maths Grade 8, Technology Grade 8, Natural Sciences Grade 8, Life Sciences Grade 10, and Physical Sciences Grades 10–12. There are 32 learners in Susan's Grade 10 class that formed the focus of this study, with 17 girls and 15 boys.

Thabo is a Physical Science and Economics teacher in a quintile five school in Gauteng province that has a population of 1120 learners. He is 45 years old. He holds a Masters in Physics and an Electronic Science degree, which he obtained in 1996. He has taught Physical Sciences, Applied Maths, Integrated Sciences, Economics, and Mathematics. He

has 23 years' teaching experience. His school does not have all the laboratory equipment, but they have some lab apparatus. Teachers have their laptops and projectors for teaching. The learners have smart phones but for this research they used four tablets to work in groups. The school was founded 32 years ago. His major subjects are Physical Sciences and Electronics Science. His Grade 10 class has 28 learners, of which 10 are boys and 18 are girls.

All three classes were mixed ability groups.

### 3.3. Data Collection and Analysis

Data was collected by means of lesson observations and interviews. Two lessons were observed per teacher, and the teachers were interviewed individually. The researcher analyzed the transcribed data by means of coding [34]. Coding is a research method that a researcher uses when interested in using an entire dataset to identify underlying themes presented through the data [35]. Codes were extracted directly from the raw data as teachers exhibited the pedagogical actions. Codes that were correlated were grouped together to develop categories, then different themes emerged from the different categories and assertions were made [36]. Eight themes on the enactment of simulations in 5E inquiry-based science teaching emerged from the lesson observations. In accordance with [37], the following strategies were used to enhance internal validity. Firstly, different sources of data were collected to confirm emerging findings. These sources included interviews, lesson plans, and class observations. There was continuous dialogue and critical reflection regarding the research process and tentative findings as they emerged. Finally, rich descriptions of events were provided.

## 4. Findings

Six 5E lessons were observed and video recorded. The topics that were covered were series and parallel circuits and the total equivalent resistance in a series and a parallel circuit. The simulation that was used can be accessed through this link: https://phet.colorado.edu/en/simulations/circuit-construction-kit-dc (accessed on 25 July 2022). All classroom activities that took place where centerd on the simulations. The teacher facilitated class discussions that arose as a result of students' involvement in the simulated activities. Below is a discussion of the themes that emerged from the analysis of the lesson observations. The themes are clustered according to the each of the 5E phases. Each theme addresses a pedagogical action undertaken by the teacher in a phase on the 5E lesson.

### 4.1. Engage Themes

The themes related to this phase suggest that the teacher gets learners to reveal their prior knowledge by getting them to build a representation of this knowledge by building and arrangement in the simulation. Teachers also used a video as a precursor to the simulation in order to stimulate interest in the topic.

4.1.1. Theme 1: Teacher Elicits Responses That Uncover Students' Prior Knowledge

The pedagogical action that Peter exhibited in the engage phase of the 5E instructional model was to elicit the prior knowledge of the learners using the PhET simulation. The teacher recognised the learners' prior knowledge by tapping into what the learners knew about the topic. Learners had to express their understanding of the topic by constructing a simple circuit to present it to their teacher as evidence of their understanding. Both Peter and Susan asked the learners the difference between a series circuit and a parallel circuit. See Peter's lesson excerpt below that refers to this engagement with learners.

> So maybe, as a first, I'm going to ask you. What does series mean? If you hear the word series, what comes to your mind? Ok let's do this . . . quickly create a series circuit in your simulations to show that you understand the meaning of a series circuit. That will take you less than five minutes. Let's go. (Peter, Lesson 1)

The pedagogical action that Thabo presented in the engage phase was to ask a question to establish the learner's knowledge of apparatus. The teacher wanted to know if the learners are familiar with the apparatus that they will use in the explore phase of the 5E instructional model. See the excerpt below.

Its potential difference, its current, and its resistance. Those three are very important just to get some feedback from you. What is a resistor? (Thabo, Lesson 2)

In all three classes, the learners' responses to the teacher questioning revealed that learners possess misconceptions of electric circuits. One of these misconceptions was that electric current travels around the circuit in one direction and that along the way the current decreases. This misconception is referred to as the attenuation model (McDermott and Shaffer).

### 4.1.2. Theme 2: Activities That Generate Interest in Learning

All three teachers used an audio-visual medium as a precursor to simulation. This was a way of generating learners' interest and preparing learners for the lesson. The videos that the teachers played made the learners enthusiastic and interested in the topic. See below excerpts from Susan's and Thabo's lessons that support this statement.

... Alright. We are going to start by a short video on how to do parallel calculations, then you are going to the simulation on your ... tablet or PC, or laptop that you have got with you to design your own circuit and then with that parallel combination of your own design, you are going to calculate equivalent resistance of that circuit. Ok. Here we go. (Thabo, lesson 1)

I am going to play a video for series and parallel circuit as an introduction to my lesson. [Teacher plays a video] (Susan, Lesson 2)

Learners watched the video for a few minutes, then Susan directed them to the simulation activity.

Alright, that is the basic explanation of the difference between a series circuit and a parallel circuit. We are now going to investigate other aspects of a series circuit and parallel circuit using a simulation.

A similar strategy was used by Peter with his class.

### 4.2. *Explore Themes*

In this phase the teachers encouraged learners to explore their initial ideas about series and parallel circuits by investigating through a hands-on simulated activity. Further to this, the teachers asked learners to work in groups on the activities.

### 4.2.1. Theme 3: Teachers Promote Learning through Hands-On Simulated Activities

The common pedagogical action among the three teachers was to ask learners to construct a circuit using PhET simulations. The teachers in this phase encouraged learner involvement through the manipulation of materials using PhET simulations. This gave the learners an opportunity to test their hypothesis and also to develop their understanding of the phenomenon that was investigated. See extracts from the three teachers' lessons that support this statement, below:

Now what I want you to do now using the PhET simulation that we have used in the current lesson I want you to construct a parallel, a parallel circuit. Right, in your parallel circuit, after you construct. We have two variables that we talked about yesterday. Eh one of the variables is current and the other variable is what? Potential difference. (Peter, Lesson 2)

In your group you choose what you want to build. So, you build a nice little circuit. With as many parallel parts as what you choose in a group, right. (Susan, Lesson 1)

> You are now going to use a simulation to first draw a series a circuit . . . Then you are going to take readings of the current that flows through different points in a circuit. (Thabo, lesson 2)

4.2.2. Theme 4: Teachers Used Questioning Strategies to Support Learning in Being Cognizant of What They Were Doing

Questioning strategies were employed by all three teachers in getting their learners to be more metacognitive of their actions. The following are excerpts to support this theme. In the excerpt below, Susan in lesson two asks a question to check their calculation.

Alright. What have we got here? have you calculated your total resistance? How have you got your 3.3 ohms?

That was your calculator method. What was your voltmeter reading? Voltmeter reading for the whole circuit.

Alright you can use that, the V over I. What is your answer? Still 3.33 ohms. So that proves that the method is correct.

*4.3. Explain Phase*

The theme below underlines that teachers afforded learners opportunities to demonstrate their understanding of the phenomenon being investigated by asking them to explain their observations and findings.

4.3.1. Theme 5: Teacher Encourages Students to Explain Their Observations and Findings in Their Own Words

Peter and Susan asked learners to present and explain their findings in groups. After collecting data in groups using PhET simulations, the learners were given the opportunity to present their findings and observation. Teachers asked learners to present their simulations to the class and explain their findings. See Peter's extract that indicates this below:

> What I want you to do now is to choose one person to . . . who will come and present their findings. You are allowed to explain using your simulations to support your statement because we don't want you to lie to us. (Peter, lesson one)

Susan in lesson two asked the learners to explain their findings to check their understanding. As the learners were presenting their findings, Susan realised that the learners had misconceptions. The teacher addressed the learner's misconception. This is reflected in the exchange below.

Susan: But what have you discovered about potential difference in a parallel connection and what have you discovered about current in a parallel circuit?

Group 1 presenter: Ok uhm, we noticed that current in a parallel circuit uhm is divided uhm more than once. It has more than one pathway of flowing. And the current that is in the circuit is dependent on the cells that produce uhm the amount of energy transmitted and the energy that the resistors has is also dependent on the cells and potential difference is divided amongst the, all the different types of components.

Susan: Right, uhm, I just want to make a small correction on potential difference. We cannot have current being divided and potential difference being divided. So, there's one of those is not correct and is going to be corrected by one of the groups. So, uhm, after you have all presented, I am going to give you conclusion of what's true and what's not true. So, my next group is the group on my left here. Who's presenting in the group here?

Thabo did not question the learners in groups, but instead identified individual learners to answer his qualitative questions that probed their understanding.

*4.4. Elaborate Phase*

In the elaboration phase, teachers facilitated a discussion on the knowledge learners had acquired by extending this learning to a broader context so that learners could apply this knowledge, and thereby achieve a deeper understanding of the phenomenon.

4.4.1. Theme 6: The Teacher Encourages Students to Apply or Extend the New Concepts

Thabo gave the learners an opportunity for students to study the main concept in a deeper or broader context. He did this by asked them to consider how the light bulbs could be connected to yield maximum brightness. See Thabo's lesson excerpt that indicate this below:

> . . . for example, I'm going to use simple example, you want light bulbs to shine the brightest, how are you arrange them? . . . How are you going to connect resistors? Eh you want a certain device, certain devices uhm to be connected and work at certain power rays. Power is not something a problem that you are going to talk about. But you want it to be like that. How are you going to connect resistors? It is important to note that circuits are everywhere. They can be found in your phone, in your watch, television, etc.as I go back to my question. you want light bulbs to shine the brightest, how are you arrange them? (Thabo, lesson two)

Similar activities were provided by Susan and Peter in extending their learners to apply what they had learned.

*4.5. Evaluate Phase*

In this phase, the teachers asked open-ended questions to establish the extent to which learners had achieved the learning goals. They also administered problems that required learners to solve quantitatively by doing calculations.

4.5.1. Theme 7: Teachers Asks Questions to Establish Learner Understanding

For example, Susan asked open-ended questions based on the simulation activity that the learners did under the Explore phase of the 5E instructional model to determine students' level of understanding. This pedagogical action allowed students to reason, reflect, and draw conclusions based on the simulation activity they have explored. See the exchange below which indicates this:

Susan: So, what did you notice about potential difference?
Learners: It's different.
Susan: Different? In what way? In regular amount? Is there . . . a . . . can you draw a conclusion in the amount of difference? If you had three light bulbs and the potential difference was nine volts, what was each potential difference?
Learners: 'three'.
Susan: The potential difference was three. So, what are we then saying about potential difference in a series circuit?

4.5.2. Theme 8: Teacher Gives Learners Quantitative Problems to Solve

Peter and Thabo gave learners quantitative problems to solve as post-lab assessments to track student gains based on the simulation activity. See Peter's lesson one excerpt below where he describes the problem for learners to solve.

> We have a battery with a total voltage of 12 ohms, and two resistors. One is a 5 ohms resistor and the other one is 15 ohm resistor and a closed switch. The question is calculate the potential difference across the 5 ohm and the potential difference across the 15-ohm resistor. So, you may discuss in your group to come with an answer. (Peter)

Thabo, in lesson two, also gave learners a post-lab problem-solving activity to measure the mastery of the learning objectives or understanding of the main concept. Similarly, Thabo prompts learning into solving a problem as follows:

> Right, so uhm, I will give you a quick exercise. A series connection is easy, so I will give you an exercise on a parallel connection. Addition of resistance. That's what I want you to, to do using this formula. So, I'm not going to erase this

formula [pointing the parallel formula for the total resistance]. It's going to be there. What I'm going to do is to, draw a circuit, and I want you to calculate the total resistance. [teacher drawing the parallel circuit]. Right, I want you to calculate the total resistance of that circuit.

Susan, in lesson two, allowed students to make a prediction and asked them to check their prediction using a simulation. The teacher has used a self-check activity whereby the learners made a prediction and used the simulations to check their answers. This falls under the evaluation phase because the learners were given the self-check activity after exploration. See Susan's lesson excerpt that supports this statement below:

I want to . . . to predict the potential difference across the battery in the parallel circuit if . . . if you are given V1 as 12. What will V1 be? Please have a reason at the back of your mind. (Susan)

## 5. Discussion

The findings of this study have shown that the teachers were able to enact appropriate pedagogical actions in facilitating learners' experiences in simulated 5E inquiry. Some of these actions are now discussed in relation to other studies that have been conducted in this area.

A notable pedagogical action that was demonstrated by the teachers in the engage phase was to elicit the students' prior knowledge using a PhET simulation. The teachers were able to do this by giving the learners a task to create a simple circuit in the PhET simulation. Norwood [30] suggested that in this phase, the teacher should check the learner's prior knowledge or any misconceptions that they might be holding. Therefore, it is important for teachers to give the learners a small activity to tap into what the learners already know about the topic. The teachers were able to effectively do this using the PhET simulation.

However, in some lessons, the teachers used a short video as a precursor to simulation. This was a way of generating learners' interest and preparing learners for the lesson. Bybee [29] says that in this phase the teacher draws the students' interest and stimulates their curiosity. A study by Chitman-Booker and Kopp [22] shows that when a teacher begins a lesson with a classroom energy-enhancing exercise, students' emotions are intensified, and they become excited and interested in the topic. This suggests that before learners are immersed in the simulation, it be viable to first stimulate their curiosity by playing them a short video on the topic.

The most common pedagogical action that was adopted by the three teachers in the explore phase was to give the learners an opportunity to be hands-on through the manipulation of materials using PhET simulations. In this study, being hands-on enabled the learners to test their hypotheses and also to improve their understanding of the phenomenon studied. This is in keeping with the assertion by Yelland [38] that simulations enable learners to engage with powerful ideas and conduct explorations that are not usually possible in classrooms. As is the case with a practical activity in a laboratory, learners' exploration of the phenomena in electricity enabled them to connect 'hands-on' and 'minds-on domains' [39]. This finding is similar to that of a study on inquiry-based learning by Chen [40], which found that an inquiry-based teaching and learning strategy had a positive impact on the growth of students' engagement, hands-on skills, and collaboration.

The teachers also encouraged learners to work collaboratively in a simulation activity. This affordance of the simulations is recognized by Zulfiqar et al. [41] who state that the use of simulations provides a collaborative environment that enhances student learning experience. Other studies have also revealed the benefit of collaborative learning during simulations. For example, Cetin [42] found that simulation-based collaborative learning positively affected students' physics achievements.

In the explain phase, teachers asked learners to explain their findings and describe their observations through the use of simulations. By doing so, the learners expressed their experiences of engagement and discovery and showed their conceptual understanding [3].

Through prompting by the teachers, the learners seamlessly re-enacted key aspects of their inquiry through a demonstration to the rest of the class. They were able to demonstrate to the teacher and their peers the significant concepts clearly. In this way, students were able to express their understanding to the teacher and their peers by referring to the PhET simulation.

In the elaborate phase of the 5E, the teachers gave the learners an opportunity to study the main concept in a deeper or broader context using a simulation. The teachers allowed the learners to link what they have learnt in the classroom with challenges they encounter in their everyday lives. This echoes Linn and His [43] who said that ICT encourages lifelong learning to allow learners to link the challenges they encounter in their lives with the material they study in the classroom.

The pedagogical action adopted by teachers in the evaluate phase was to allow learners to do calculations and check their answers on the simulations. Podolefsky et al. [11] have argued that PhET simulations offer a high degree of interactivity in terms of user control and dynamic feedback. Therefore, for assessment purposes, teachers can use self-check activities that allow learners to do calculations and check their calculated values in the simulation to track their progress.

Another pedagogical action that the teachers employed in the evaluate phase was to ask open-ended questions based on simulation activities to determine students' level of understanding. It is important for teachers to ask students questions based on activities carried out in a simulation to promote high-order thinking, reflection, and the drawing of conclusions. A study conducted by Perkins et al. [13] also found that a simulation can be used to reinforce ideas and to help students reflect on what they have learnt. Therefore, it is important to promote student reflection of what they have learnt using simulations through open-ended questions.

## 6. Conclusions

The findings of this study show that PhET simulations can be used to good effect in supporting leaners in doing 5E inquiry in a meaningful manner. All three teachers were able to facilitate learner engagement with the simulations to support conceptual understanding, collaborative learning, communication of results, and to get learners to support the claims they made based on simulation activity. The teachers were able to effectively scaffold learner engagement across each of the phases in 5E. The 5E model is based on the constructivist theory to learning, which suggests that people construct knowledge and meaning from experiences. Through the use of the simulated activity teachers were able to support leaners to reflect on activities to reconcile their new knowledge with previous ideas. While it is acknowledged that the PhET simulated activity is not a substitute for hands-on practical work in a laboratory, it can be a powerful tool for supporting learning. The interactivity offered to learners in the simulated activity enabled teachers to scaffolding learners in concept formation. This in contrast to other traditional teacher-centerd approaches that have not had the desired effect in supporting learners acquire an understanding of electric circuits [44].

A limitation of the study was that the sample size was only three teachers. This could possibly constrain any attempt at generalizing the findings to other school settings.

While the scope of this study was confined to how the teacher enacted the simulations across the 5Es, it recommended that future research can be planned to focus on how learners experienced the simulations in 5E inquiry.

**Author Contributions:** Conceptualization, G.M. and U.R.; methodology, G.M. and U.R.; software, G.M. and U.R.; validation, G.M. and U.R.; formal analysis, G.M. and U.R.; investigation, G.M.; resources, G.M.; data curation, G.M.; writing—original draft preparation, G.M. and U.R.; writing—review and editing, U.R.; visualization, G.M and U.R; supervision, U.R.; project administration, G.M. and U.R.. All authors have read and agreed to the published version of the manuscript.

**Funding:** This research received no external funding.

**Institutional Review Board Statement:** The study was conducted in accordance with the Declaration of Helsinki, and approved by the Institutional Review Board (or Ethics Committee) of the University of Johannesburg (protocol code 2019-069 on 17 October 2020 and date of approval).

**Informed Consent Statement:** Informed consent was obtained from all subjects involved in the study.

**Conflicts of Interest:** The authors declare no conflict of interest.

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
