# Peer review of "Physical Sciences Teachers’ Enactment of Simulations in 5E Inquiry-Based Science Teaching"

_education, doi:10.3390/educsci12120864_

Round 1
Reviewer 1 Report
The authors have performed an interesting ethnographic study but I consider it lacks the depth of results. For example, the authors do not consider the abundant literature regarding the teaching of circuits using inquiry-based learning and/or PhET simulations.
Here are some comments to help improving the manuscript:
The science topics to be examined in the research should be briefly introduced and some of the expected misconceptions or predicted difficulties briefly explained. What are the problems when teaching circuits? What are the best experiences? Is there any research on 5E teaching for the topic of circuits? (I believe there is abundant literature)
It may not be necessary to explain in such detail the 5E cycle of inquiry, since it has been thoroughly explained by Bybee, Pedaste and others.
METHODS
Line 169-Give a reference for the meaning of five quintile school
Show the videos shown by the teachers and a link to the PhET simulation
The authors should explain whether other type of classroom activities to introduce topics, formula, etc were used before, during or after the PhET simulation activity.Limitations of the work are nearly missing. Comparison with other methodologies to teach circuits would have been a bonus to argue for the method employed by the three teachers.
MINOR COMMENTS
LINE 31. Substitute "emphasise" for "emphasises"
Line 37. Include bracket ( before "Harlen, 2013)".
Line 50. Remove bracket before the date in "(Almasri (2022)"
Line 54. Remove "is" before "commonly"
Line 55. Insert "," after "simulation"
Line 59. Remove "," after Perkins in "Perkins, et al., 2006,1)
Line 77. Remove space between "and" and "pedagogical"
Line 94. Add space after & in "(Duran &Duran, 2004)"
Lines 94-96. Use of "because" twice in the same sentence. Rephrase it.
In Figure 1, one of the phase is called Extension, but authors use Elaboration instead. Decide which term you want to use
119. Correct "suggest" for "suggests"
Line 146. Include "," after "phase"
Line 194. Remove "s" in " per teachers"
Line 272. "support" repeated twice. Maybe the authors mean "theme"?
REFERENCES
check the format of the reference list
Author Response
Attached find a document that details the changes made.

Reviewer 2 Report
Review Report
Article title:
Physical Sciences teachers’ enactment of simulations in 5E inquiry-based science teaching
This case study investigated South African Physical Sciences teachers’ enactment of simulations in 5E inquiry-based science teaching at three high schools.
The authors made a significant effort to guide the reader through the overall research study. The problems of this study are detailed below.
Specific comments:
1. The abstract of the manuscript should contain a clearly formulated objective of the case study.
2. The form of the research question should be: “How?” “Why?”
3. Specify what was the main criterion for selecting the research sample.
4. A case study is a research approach that is used to generate an in-depth, multi-faceted understanding of a complex issue in its real-life context. It is an established research design that is used extensively in a wide variety of disciplines, particularly in the social sciences. A case study can be defined in a variety of ways, the central tenet being the need to explore an event or phenomenon in depth and in its natural context.
a) A case study should focus on a detailed description of each case.
b) A learning objective should be listed for each theme.
c) The case study was concerned with the application of PhET simulations in the science teaching. It would be helpful to specify the concrete PhET simulation that was used in the class for each theme.
5. At the end of the discussion, the case study should be placed in a wider context. An assessment of the validity of the results should also be made.
6. Arthur Eisenkraft elevated the 5E model to the 7E model (Engage/Elicit, Explore, Explain, Elaborate/Extent, Evaluate) based on his research (2003). The proposed 7E model expands the engagement element into two components: Elicit and Engage. Similarly, the 7E model expands the two stages of elaboration and evaluation into three components: Elaborate, Evaluate, and Extend. The goal of the 7E learning model is to emphasize the increasing importance of eliciting prior understandings and the extending, or transfer, of concepts. With this new model, teachers should no longer overlook these essential requirements for student learning. Can the authors explain why they used the 5E and not the 7E model in their study?
7. References must be numbered in order of appearance in the text and listed individually at the end of the manuscript. In the text, reference numbers should be placed in square brackets [ ] and placed before the punctuation.
8. References are not formatted in accordance with the guidelines for authors.
Author Response

(The authors gave the same response as above.)

Reviewer 3 Report
Comments and Suggestions for Authors
This manuscript aims at investigating “South African Physical Sciences teachers’ enactment of simulations in 5E inquiry-based science teaching at three high schools” (abstract, line 4-5) and “the research was guided by the following question: How do Physical Sciences teachers enact simulations in 5E inquiry-based science teaching?” (line 80-81). Although the issue investigated in this study could be of some interest to the readers of Education Sciences, there are some criticalities that I would like to highlight.
· “How do Physical Sciences teachers enact simulations in 5E inquiry-based science teaching?” (line 80-81) appears to be a general purpose, which differs from a research question. In this study it seems that the purposes of simulations employment could be both to engage the students and to bring out some of their possible misconceptions about electric circuits. The authors may want to explicitly list research questions in order to favour the readers’ comprehension of the results and the conclusions.
· The bibliography does not appear on the whole to be current. Less than 25% of the cited references were published within the last five years. The authors may want to implement some recent and appropriate papers.
· The Methods section should be expanded.
a. In the Profile of teachers paragraph, the authors illustrate some features of the three teachers involved in this research. It could be useful to know their age. Furthermore, their principal charactheristics could be summarised in a table.
b. In this section there is not a paragraph which describes the students characteristics. How many students were involved in each of the three schools? What was their age and gender? It seems that they were arranged in some different groups; how many groups were formed and what was their composition? And how were they grouped?
c. It is difficult to understand how each teacher developed the five elements (Engagment, Exploration and so on) required by the model. Furthermore, in the paragraphs concerning their description (from line 213 to line 373) there is not a systematic comparison of all the teachers. For instance, in the paragraph Theme 1: Teacher elicits responses that uncover students’ prior knowledge, Susan is not mentioned and the same happens in other paragraphs. The author may want to implement them.
d. The authors did not illustrate in details the students’ prior conceptions about the topic addressed; consequently, it is not possible to evaluate the effectiveness of their strategy in order to overcome some possible students’ misconceptions. More generally, the authors should describe and compare each student group.
· The Discussion section is not a discussion section. The authors describe what the teachers did during the lessons in term of the five elements of the model. But the key point is if their use of the simulations was effective regarding the aims which they wanted to achieve and they should refer to the results presented in the previous section. The authors may want to expand this section.
· The conclusion section cannot be evaluated if the discussion one does not discuss the results of the study.
Author Response

(The authors gave the same response as above.)

Round 2
Reviewer 1 Report
Dear authors and editors,
The authors have thoroughly addressed the reviewers' comments and have improved the manuscript considerably.
I am including a short comment to warn of the importance of mantaining coherence within the text when addressing our comments.
Other than that, I see the manuscript as fit for consideration. Below are my comments:
2nd revision
Lines 74-75
After the research question, the authors have included a short paragraph regarding expected misconceptions about circuits.
It is ok to follow the reviewer's advice but the coherence within the text should be maintained.
Why is it important to mention misconceptions? What is the role of making misconceptions arise in a constructivist approach such as the 5-E inquiry model? Please, reviese the theoretical background of the 5e model and justify the need to use student's ideas during the inquiry cycle.
New paragraph in Lines 101-105.
Line 102. Correct "reveal" for "revealed".
Line 103. Correct "learner" for "learners'".
Line 104. Include "a" between "provided" and "better".
Line 105. Include "and" before "created".
Author Response
Attached please find responses in the attached file.

Reviewer 2 Report
Dear authors,
congratulations on improving your manuscript. You have significantly improved the clarity of your writing and have addressed most of my concerns.
Kind regards,
The reviewer
Author Response
Thank you for acknowledging that we have addressed your comments.
Reviewer 3 Report
With relation to my previous comments, I wish to point out that the authors answered positively to some critical points which I highlighted and the changes in the paper are pertinent.
Author Response

(The authors gave the same response as above.)
